# Landscape Connectivity Limits the Predicted Impact of Fungal Pathogen Invasion

**DOI:** 10.3390/jof6040205

**Published:** 2020-10-03

**Authors:** Zhimin Li, An Martel, Sergé Bogaerts, Bayram Göçmen, Panayiotis Pafilis, Petros Lymberakis, Tonnie Woeltjes, Michael Veith, Frank Pasmans

**Affiliations:** 1Wildlife Health Ghent, Faculty of Veterinary Medicine, Ghent University, Salisburylaan 133, B9820 Merelbeke, Belgium; Zhimin.li@ugent.be (Z.L.); an.martel@ugent.be (A.M.); 2Lupinelaan 25, NL5582 CG Aalst, The Netherlands; s-bogaerts@hetnet.nl; 3Faculty of Science, Department of Biology, Zoology Section, Ege University, TR-35100 İzmir, Turkey; 4Department of Zoology and Marine Biology, School of Biology, National and Kapodistrian University of Athens, Panepistimioupolis, Ilissia, 15784 Athens, Greece; ppafil@biol.uoa.gr; 5Natural History Museum of Crete, School of Sciences and Engineering, University of Crete, Knossos Ave., 1409 Irakleio, Greece; lyberis@nhmc.uoc.gr; 6Molenweg 43, 6542PR Nijmegen, The Netherlands; a.woeltjes7@upcmail.nl; 7Department of Biogeography, Trier University, Universitätsring 15, D-54296 Trier, Germany; veith@uni-trier.de

**Keywords:** *Batrachochytrium salamandrivorans*, salamander, *Lyciasalamandra*, thermal ecology, susceptibility

## Abstract

Infectious diseases are major drivers of biodiversity loss. The risk of fungal diseases to the survival of threatened animals in nature is determined by a complex interplay between host, pathogen and environment. We here predict the risk of invasion of populations of threatened Mediterranean salamanders of the genus *Lyciasalamandra* by the pathogenic chytrid fungus *Batrachochytrium salamandrivorans* by combining field sampling and lab trials. In 494 samples across all seven species of *Lyciasalamandra*, *B. salamandrivorans* was found to be absent. Single exposure to a low (1000) number of fungal zoospores resulted in fast buildup of lethal infections in three *L. helverseni*. Thermal preference of the salamanders was well within the thermal envelope of the pathogen and body temperatures never exceeded the fungus’ thermal critical maximum, limiting the salamanders’ defense opportunities. The relatively low thermal host preference largely invalidates macroclimatic based habitat suitability predictions and, combined with current pathogen absence and high host densities, suggests a high probability of local salamander population declines upon invasion by *B. salamandrivorans*. However, the unfavorable landscape that shaped intraspecific host genetic diversity, lack of known alternative hosts and rapid host mortality after infection present barriers to further, natural pathogen dispersal between populations and thus species extinction. The risk of anthropogenic spread stresses the importance of biosecurity in amphibian habitats.

## 1. Introduction

With half of all extant species of salamanders and newts (urodeles) estimated to be threatened or extinct, this amphibian order is among the globally most threatened vertebrate taxa (Global Amphibian Assessment, 2008). The recent incursion of the chytrid fungus *Batrachochytrium salamandrivorans* in Europe added a threat to the survival of most western Palearctic urodele taxa [1]. *B. salamandrivorans* induced urodele die offs now have been diagnosed in four European species in four countries. Invasion of *B. salamandrivorans* in populations of endangered, small range endemics may lead to species extinction [2]. Predicting the impact of *B. salamandrivorans* invasion in threatened urodele communities is key to guiding conservation priorities and is often based on macroclimate niche modeling and/or experimental infections [3,4,5,6,7,8]. Neglecting host dependence on specific microclimatic conditions may, however, confound such risk analyses.

The eastern Mediterranean is characterized by a remarkable diversity of highly specialized, terrestrial salamanders of the genus *Lyciasalamandra*. This genus contains 7 species and 20 subspecies, all of which are threatened according to International Union for Conservation of Nature (IUCN). Most of these (sub-) species occupy small ranges and are restricted to habitats that provide access to underground, relatively cool and humid conditions year round. This allows them to escape the arid and hot summer conditions [9,10], but also unfavorable weather conditions during the winter [11]. Surface activity of these salamanders is restricted to cool and wet weather and at least for coastal populations, most activity is between November and April [12]. Their close relatedness to salamanders that are considered highly vulnerable to *B. salamandrivorans*-induced disease (European fire salamanders, *Salamandra salamandra*; further: fire salamander), low reproductive output (females typically give birth to two fully developed offspring once a year), relatively low life expectancy [11,13,14] and small distribution ranges could prompt conclusions that *Lyciasalamandra* species are highly vulnerable to severe, infectious disease driven declines and even extinction. The extent of the impact would, however, also be determined by pathogen transmission success between salamander populations and by host thermal ecology. *B. salamandrivorans* has been shown to spread slowly [15], and natural (non-anthropogenic) spread is associated with the presence of pathogen carriers [2,16,17]. Salamander body temperatures that exceed the lethal temperatures of 25 °C for *B. salamandrivorans* could suppress or even eliminate infection [18,19]. We here predict the risk *B. salamandrivorans* poses to the survival of the genus *Lyciasalamandra* by combining a susceptibility lab trial with a *B. salamandrivorans* prevalence study in the species’ range and collecting lab and field data on the host species‘ thermal ecology.

## 2. Materials and Methods

### 2.1. Infection Dynamics of B. Salamandrivorans in Lyciasalamandra Helverseni

We compared infection dynamics of *B. salamandrivorans* between *Lyciasalamandra helverseni* and the reference urodele species for susceptibility: the fire salamander (*Salamandra salamandra*). Permits were obtained to transfer *Lyciasalamandra helverseni* from the Greek island of Karpathos to captivity (149621/3441/28-12-2016). For conservation reasons, an absolute minimum of three replicates was used, which should suffice in case the animals show an infection pattern broadly similar to that of the well-studied fire salamanders. Three captive, subadult *L. helverseni* and three captive bred, subadult fire salamanders were inoculated with 10^3^ zoospores of the *B. salamandrivorans* type strain AMFP1 using an established protocol [1]. Briefly, animals were individually exposed for 24 h at 15 °C to 10^3^ zoospores of *B. salamandrivorans* in 1 mL of water. Afterwards, the salamanders were housed individually in plastic containers, lined with moist tissue and containing a polyvinyl chloride (PVC) pipe as a hiding place. Temperatures were kept constant at 15 °C and dim, natural light was provided. The salamanders were fed two times weekly ad libitum with crickets (*Acheta domestica*). Infection loads were followed up by weekly sampling of all animals using cotton tipped swabs and subsequent qPCR analysis to quantify *B. salamandrivorans* loads [20]. The experiment was approved by the ethical committee of Ghent University (2017/75) on October 30th, 2017.

### 2.2. B. Salamandrivorans Prevalence in Populations of the Genus Lyciasalamandra

Across the range of all 7 species of *Lyciasalamandra* (Figure 1), skin swabs were collected during the surface activity period of the salamanders in Turkey (winter, spring of 2013) and on the Greek islands of Kasos and Karpathos (March 2017). Animals were found resting under stones during daytime or were found active at the surface at night. Animals were caught by hand. For each animal, a new pair of disposable vinyl gloves was used. Animals were released on site immediately after sampling. A minimum of 30 specimens per *Lyciasalamandra* species was envisaged. This allows the detection of a *B. salamandrivorans* prevalence of at least 10% with 95% confidence. To obtain a Bayesian 95% credible interval for prevalence, the R2WinBUGS package and WinBUGS were used to estimate the posterior distribution of prevalence following the description of previous studies [21,22]. We used a uniform prior probability distribution for prevalence estimates (e.g., prevalence~U). Three parallel Markov chains with 20,000 iterations each were run, discarding the first 5000 iterations as the burn-in. Chains were not thinned.

### 2.3. Thermal Ecology of Lyciasalamandra Sp

We here assessed to what extent the *Lyciasalamandra* body temperatures overlap with the *B. salamandrivorans* thermal envelope (spanning 5 °C–20 °C, [23]). We collected two sets of field data to estimate *Lyciasalamandra* body temperatures. A first set of data consisted of opportunistically collected data of Turkish populations during the species’ period of surface activity (winter and spring) from 1996–2020 (Appendix A). During March 2017, temperature data were collected from a larger sample of *L. helverseni* from Kassos and Karpathos. Temperatures were measured at a maximum distance of 30 cm using an infrared thermometer (Fluke 561). To determine thermal preference of *L. helverseni*, 10 salamanders were placed in a thermal gradient container (80 cm × 19 cm) with temperatures ranging between 5 °C–35 °C on moist tissue, with extensive access to shelters across the thermal range. Salamanders were housed in groups of 3 or 4 animals and left to acclimate for 2 days before the onset of measurements. Body temperatures were measured during the following 3 days using the infrared thermometer pointed at midbody, at a distance of less than 5 cm from the animal. At least 6 temperature readings per animal were recorded.

Statistics were performed using SPSS version 25 (SPSS Inc., Chicago, IL, USA), by performing a non-parametric one-way ANOVA analysis on the temperature, with significance set to *p* < 0.05.

### 2.4. Thermal Environment of B. Salamandrivorans in Lyciasalamandra Habitats

Since *B. salamandrivorans* poorly tolerates elevated temperatures and is eliminated from urodele hosts within 10 days (Blooi et al., 2015), the host thermal environment may facilitate predicting the risk and impact of *B. salamandrivorans* invasion. The seasonal temperature variation at 15 Turkish *Lyciasalamandra* sites, comprising 6 species (Appendix A), was recorded using Tinytag Dataloggers (Gemini Data Loggers UK Ltd., Chichester, UK.) between February 1997 and February 1999. For comparison between sites, we recorded the local temperature under big rocks (typical daytime shelters during surface activity); measurements were taken at the respective end of three-hour intervals throughout almost one year per site and logger. In addition, at some sites we also studied potential salamander hiding places (within stone walls, mammal holes, leaf litter and boulder fields) in order to see how temperature variations are buffered within underground shelters. To account for the thermal critical maximum of 25 °C of *B. salamandrivorans* [23], we counted the number of days with a temperature exceeding 25 °C and the number and longest period of days where the temperature did not drop below 25 °C.

## 3. Results

### 3.1. Lyciasalamandra Helverseni Quickly Develops Lethal Infection Loads of B. Salamandrivorans

Experimental infection of *L. helverseni* resulted in rapid increase in *B. salamandrivorans* loads (average of log (10) genomic equivalents (GE): 4.32 +/− 0.31 (standard deviation)) and humane end points for euthanasia (lethargy, skin lesions, abnormal postures) were reached at 4 weeks post exposure in all three animals. Infection dynamics and disease course were highly comparable to those observed in the reference fire salamanders (Figure 2).

### 3.2. B. Salamandrivorans Is Absent from Natural Populations of Lyciasalamandra sp

In total, 494 samples of *Lyciasalamandra* species across the entire genus range were found negative for *B. salamandrivorans* (Table 1). For each species, at least 30 samples were examined (Appendix A).

### 3.3. Lyciasalamandra Body Temperatures Fit the B. Salamandrivorans Thermal Envelope

All temperatures measured during salamander surface activity in the field and in the lab trial are well below the thermal critical maximum of 25 °C of *B. salamandrivorans* [23]. The opportunistic sampling of Turkish specimens yielded 19 measurements from five species. Animals in hiding places had an average body temperature of 12.8 °C (range: 8.4 °C–16.4 °C, *n* = 12) and active animals 10.6 °C (range: 6 °C–14.2 °C, *n* = 7). For the Greek *L. helverseni*, average body temperatures were 10.5 °C (range: 3.8 °C–14.6 °C, *n* = 35) in hiding places and 8.2 °C (range: 4.6 °C–12.8 °C, *n* = 75) when active (Appendix A).

*L. helverseni* housed in a thermal gradient had an average body temperature of 16.3 °C +/− 0.6 °C (standard error of the mean (SEM)). A marked group effect (*p* ≤ 0.001) was noticed, and average temperatures per group were 13.5 °C, 17.1 °C and 18.0 °C. Of 68 temperature measurements, body temperature exceeded 20 °C on three occasions, and the highest temperature recorded during the experiment was 20.5 °C.

The temperature loggers placed in the field in 15 Turkish populations yielded consistent results during the salamanders’ surface activity period (November–April; Figure 3, Appendix A).

### 3.4. Thermal Conduciveness of Lyciasalamandra Surface Habitats for B. Salamandrivorans Survival

On average, 2858 readings were collected per temperature logging device. Temperatures in alleged salamander hiding places varied year-round between −0.1 °C –41.3 °C, with an overall average temperature of 17.5 °C (SEM 0.6 °C). When considering a constant temperature of at least 25 °C over at least 10 days lethal for *B. salamandrivorans*, temperature readings at 12 sites suggested poor year-round fungal survival at the site of recording. In contrast, three sites at higher elevation did not yield a single day with temperatures exceeding 25 °C (Figure 3).

## 4. Discussion

Combined results from lab trials and field data suggest that invasion of *B. salamandrivorans* poses a threat to *Lyciasalamandra* species at population level, with expected declines as witnessed in fire salamanders across Europe [23]. Animals succumbed quickly after experimental infection with a single, relatively low dose of *B. salamandrivorans*, in line with the disease outcome observed in their sister genus *Salamandra* [1]. Within population transmission is highly likely, given the high population densities of *Lyciasalamandra* species [10], frequent communal use of hiding places [24] and courtship behavior that involves intimate contact (amplexus, [25]). Whether infection and disease dynamics during a natural outbreak would follow a similar scenario to those in fire salamanders remains unclear. The rapid decline and extirpation observed in the latter have been associated with the presence of pathogen reservoirs, which at least partly explains why the epidemiology does not follow a density dependent pattern [17]. With multiple, highly transmitting hosts in the community, transmission increases, resulting in a higher possibility of a severe outbreak [26]. Known pathogen reservoirs include co-occurring amphibian hosts that are persistently and often subclinically infected [17]. Although at least some anurans may be *B. salamandrivorans* carriers, natural infections in frogs and toads so far have been observed in eastern Asian species [27], and are very rare in anurans of European amphibian communities invaded by *B. salamandrivorans* [1,28]. Furthermore, the surface of the karstic limestone habitats inhabited by the salamanders is not home to a significant anuran community. If the lack of other urodele species and the scarcity of anuran hosts in the *Lyciasalamandra* range would imply absence of suitable pathogen reservoirs, infection and disease dynamics can be expected to be density dependent [26,29], rendering population extirpation less likely. However, current knowledge does not allow the exclusion of the presence of non-amphibian pathogen reservoirs, which may be biotic (for example invertebrate pathogen reservoirs as proposed for *B. dendrobatidis* [30,31] or abiotic (long term survival of resistant fungal spores). Abiotic reservoirs should maintain sufficient humidity levels and suitable temperatures for *B. salamandrivorans* to survive. *B. salamandrivorans* infection was shown to be transmitted through contaminated soil [17]. *B. salamandrivorans* DNA was demonstrated up to one month in contaminated soil, incubated at 15 °C, where the fungus can remain infective for at least 48 h. *B. salamandrivorans* persistence in superficial soil layers is likely during the humid season only. Our thermal recordings within the salamander habitats showed that, for the majority of sites, *B. salamandrivorans* is unlikely to survive the hot Mediterranean summers in the superficial soil layers in most habitats. In the absence of alternative hosts, desiccation of the few habitats that did fall within the *B. salamandrivorans* thermal envelope year-round can be expected to equally limit fungal survival in superficial soil layers. However, at greater depths in the karstic limestone underground, the presumed summer refugia of *Lyciasalamandra*, the consistently lower temperatures would offer suitable conditions for year round fungal survival. Given the comparable thermal niches of both *Lyciasalamandra* sp. and *B. salamandrivorans*, a persistently contaminated subterranean environment could markedly exacerbate the disease impact and may result in mass mortality events that go unnoticed.

Observations regarding thermal biology of the Mediterranean salamander genus *Lyciasalamandra* suggest a clear preference for temperatures below 20 °C. A similar preference for relatively low body temperatures has been observed in other Mediterranean urodele species inhabiting similar macroclimates [32,33,34,35,36,37]. This may be an important mechanism for survival of Mediterranean salamanders, since cooler temperatures coincide with refugia that provide sufficient humidity to survive the hot and dry summer. The predominantly karstic limestone region inhabited by *Lyciasalamandra* provides such conditions in the deeper karstic underground system. The overlap between thermal behavior of the host species and the thermal niche of the pathogen would favor growth of the latter, with the potential to cause disease and death in the host in nature, despite inhabiting superficially (for the fungus) unfavorable habitats. Predictions of suitability based on macroclimatic data and surface characteristics in such cases poorly represent, and in this case underestimate, the potential invasive range given extensive buffering of the salamanders’ niche from macroclimatic conditions. Any predictive niche modeling of infectious diseases in ectothermic hosts should therefore focus on microclimatic conditions that take host ecology into account [18].

At species level, *B. salamandrivorans* can be expected to be less likely to cause extinction. Extinction would be a function of extirpation at the population level, combined with successful pathogen transmission to neighboring host populations. The genus *Lyciasalamandra* is characterized by marked genetic differentiation even at a very small local scale [10,38]. Although a limited number of contact zones between some of these lineages do exist, the generally inhospitable landscape that shaped this genetic diversity can be expected to result in poor connectivity and marked landscape resistance against the *Lyciasalamandra* host dispersal. Assuming that salamanders of the genus *Lyciasalamandra* are the only significant hosts present, this may heavily impact the epidemiology of the infection, since dispersal of *B. salamandrivorans* has been associated predominantly with dispersal of infected hosts [15]. Ironically, rapid mortality post infection further limits pathogen spread through infected hosts. This would mean that incursion by *B. salamandrivorans* is likely to affect any of the *Lyciasalamandra* species across their entire range only in the case of human intervention.

We therefore propose that *B. salamandrivorans* could pose a significant threat to the survival of any of the *Lyciasalamandra* taxa only in the case of multiple, independent introduction events or in the case of anthropogenic spread. Currently, introduction routes of *B. salamandrivorans* can only be speculated about, but human activities are held responsible for its arrival and expansion in Europe [1,39]. A recent case in Spain at over 1000 km of the index outbreak was linked to the release of infected, captive urodeles [2]. Since *Lyciasalamandra* are strictly terrestrial, introduction of *B. salamandrivorans* through natural vectors like waterfowl [17] is unlikely. The highly diverse populations of *Lyciasalamandra* combined with the presence of excellent and extensive tourist infrastructure attract a specific crowd of (often foreign, European) tourists and scientists who often visit several populations in a short timeframe. Since fomites have been suggested to play a role in introduction and spread of the pathogen [40,41,42,43] we emphasize the importance of biosecurity during activities in urodele habitats. Handling animals should be done using disposable gloves and the use of containers to temporarily house animals should be avoided. Any materials that may come in contact with the salamanders or their environment, including footwear, should be thoroughly cleaned and disinfected before visiting a different population.

In conclusion, *B. salamandrivorans* introduction is likely to result in local population declines of *Lyciasalamandra* species that may go unnoticed in the deep limestone underground, however, with a low probability of whole range spread and species extinction if further anthropogenic spread can be prevented.

## Figures and Tables

**Figure 1 jof-06-00205-f001:**
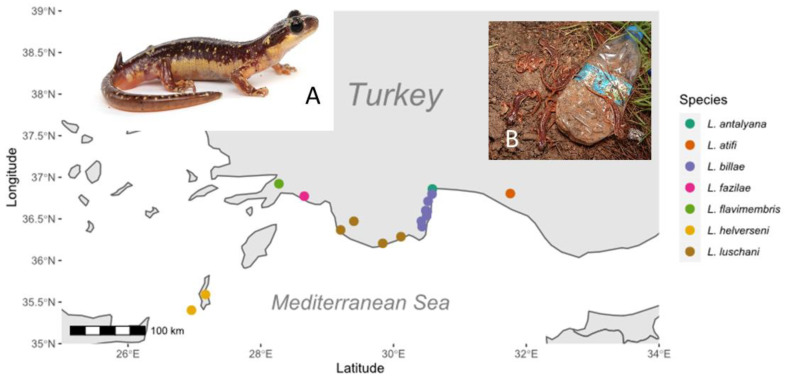
Sampling locations of the 7 *Lyciasalamandra* species in Turkey and the Greek islands of Karpathos and Kasos. The inserts show an adult male *Lyciasalamandra helverseni* (**A**) and communal use by *L. billae* of a hiding place in an anthropogenically disturbed habitat (**B**).

**Figure 2 jof-06-00205-f002:**
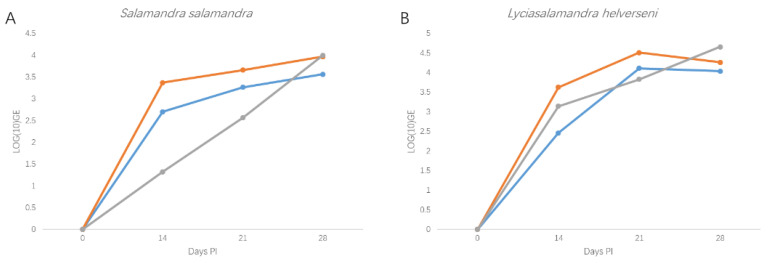
Infection dynamics in *S. salamandra* (**A**) and *L. helverseni* (**B**) after experimental exposure to *B. salamandrivorans*. Infection loads are shown as log (10) GE values. Each line represents a single animal.

**Figure 3 jof-06-00205-f003:**
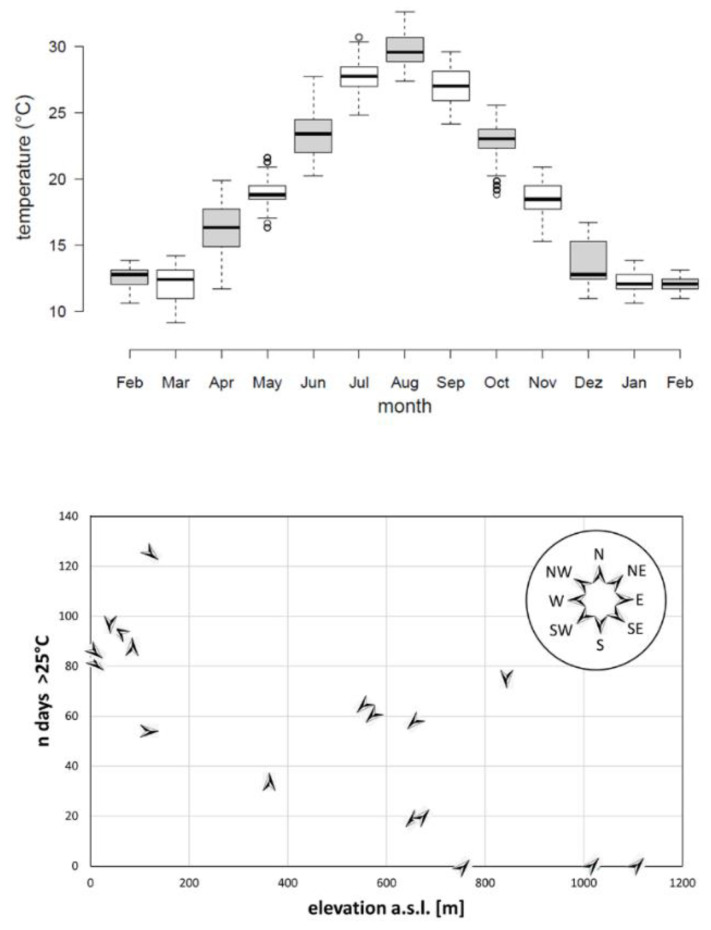
Year round temperature recordings in 15 *Lyciasalamandra* habitats. The upper panel shows box plots of temperature recordings in an alleged salamander hiding place under a stone in a representative coastal population of *L. billae* as an example of seasonal fluctuations in a population where most surface activity is between November and April. The lower panel shows the number of days on which the recorded temperature exceeded 25 °C in alleged salamander hiding places in relation to altitude. Temperatures were recorded year-round in 15 populations belonging to 6 Turkish *Lyciasalamandra* species. Arrows indicate the orientation of the slope, defined as the compass direction that the slope faces toward.

**Table 1 jof-06-00205-t001:** All seven species of *Lyciasalamandra* populations were sampled for *B. salamandrivorans* presence, the last column indicates the Bayesian 95% credible intervals.

Species	Sample Size	*B. Salamandrivorans* Prevalence	(Bayesian 95% Credible Interval)
*L. atifi*	30	0.00	(0.00, 0.11)
*L. antalyana*	30	0.00	(0.00, 0.11)
*L. billae*	97	0.00	(0.00, 0.04)
*L. luschani*	121	0.00	(0.00, 0.03)
*L. fazilae*	30	0.00	(0.00, 0.11)
*L. flavimembris*	30	0.00	(0.00, 0.11)
*L. helverseni*	156	0.00	(0.00, 0.03)
total	494	0.00	(0.00, 0.01)

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
