# Peer review of "Landscape Connectivity Limits the Predicted Impact of Fungal Pathogen Invasion"

_jof, 2020, doi:10.3390/jof6040205_

Round 1

Reviewer 1 Report

Title: Landscape connectivity limits the predicted impact of fungal pathogen invasion.

Overview:This is a well-written and important contribution to our understanding of the Batrachochytrium salamandrivorans (Bsal)infection status and susceptibility of this interesting group of salamander species. The results and conclusions are supported by the data.I have only minor suggestions for clarification and suggestions for mention of biosafety (biosecurity)protocols that could be recommended to continue to protect these threatened species.

Specific Criticisms:

1.In the methods section, please comment on the field protocols followed to assure that the scientists did not transmit Bsal infections.

2.In the discussion, please suggest guides for the tourism industry or governing bodies to prevent accidental introduction of Bsal by tourists.

3.Lines 50-54. The sentence beginning with "This genus..." is too long with too many clauses. Please break up into more than one sentence.

4.Line 56. You need a hyphen between the words "salamandrivorans" and "induced".

5.Lines 95-96. Please explain the terminology "uniform prior for prevalence".

6.Line 125. Change "three hours intervals" to "three-hour intervals".

7.Lines 175-176. It is not clear what is meant by "orientation of the slope". Is this slope of terrain? This figure (lower panel) is not perfectly clear. It could be replaced with a table.

8.Line 193. A better choice or word for "Besides" would be "Furthermore".

9.Lines 213-214. This is an important point that the subterranean environment would be suitable habitat for the fungus, and losses would go undetected

Author Response

Specific Criticisms:

  1. In the methods section, please comment on the field protocols followed to assure that the scientists did not transmit Bsal infections.

We thank the reviewer for the suggestions. For the species involved, we did not use dipnets or containers to house them. The only contact consisted of briefly handling the animals while wearing gloves during sampling. We added lines 91-92 mentioning that “Animals were caught by hand. For each animal, a new pair of disposable vinyl gloves was used. Animals were released on site immediately after sampling.”

  1. In the discussion, please suggest guides for the tourism industry or governing bodies to prevent accidental introduction of Bsal by tourists.

Mainstream tourism visiting the tourist infrastructure along the coast is unlikely to pose any significant problems. The area is, however, visited frequently by a limited number of herpetofauna oriented travelers and scientists. These people tend to visit several localities in a short timeframe. We suggest the following guidelines (line 252) that may reduce the risk of B. salamandrivorans introduction and we rephrased the sentence: “The region inhabited by Lyciasalamandra represents one of the main touristic areas of Turkey and the populations of these salamanders are frequently visited by (often foreign, European) tourists and scientists alike. Since fomites have been suggested to play a role in introduction and spread of the pathogen [40-43] we emphasize the importance of biosecurity during activities in urodele habitats.” by:

“The highly diverse populations of Lyciasalamandra combined with the presence of excellent and extensive tourist infrastructure attract a specific crowd of (often foreign, European) tourists and scientists who often visit several populations in a short timeframe. Since fomites have been suggested to play a role in introduction and spread of the pathogen [40-43] we emphasize the importance of biosecurity during activities in urodele habitats. Handling animals should be done using disposable gloves and the use of containers to temporarily house animals should be avoided. Any materials that may come in contact with the salamanders or their environment, including footwear, should be thoroughly cleaned and disinfected before visiting a different population.”

  1. Lines 50-54. The sentence beginning with "This genus..." is too long with too many clauses. Please break up into more than one sentence.

We divided the sentence into two short ones. “This genus contains 7 species and 20 subspecies, all of which are threatened according to IUCN. Most of these (sub-)species occupy small ranges and are restricted to habitats that provide access to underground, relatively cool and humid conditions year round. This allows escaping the arid and hot summer conditions [9,10], but also unfavourable weather conditions during the winter [11].”

  1. Line 56. You need a hyphen between the words "salamandrivorans" and "induced".

We changed “salamandrivorans induced” to “salamandrivorans-induced” as requested.

  1. Lines 95-96. Please explain the terminology "uniform prior for prevalence".

Using prior probability distribution is a key compound of Bayesian statistical inferences. We replaced the sentence by:

“We used a uniform prior probability distribution for prevalence estimates (e.g., prevalence ~ U (0,1).”

  1. Line 125. Change "three hours intervals" to "three-hour intervals".

We changed "three hours intervals" to "three-hour intervals" as requested.

  1. Lines 175-176. It is not clear what is meant by "orientation of the slope". Is this slope of terrain? This figure (lower panel) is not perfectly clear. It could be replaced with a table.

We would prefer to keep this figure in the manuscript but tried to clarify it by replacing the legend: “The lower panel shows the number of days on which the recorded temperature exceeded 25°C in alleged salamander hiding places in relation to altitude. Temperatures were recorded year-round in 15 populations belonging to 6 Turkish Lyciasalamandra species. Arrows indicate the orientation of the slope, defined as the compass direction that the slope faces toward.”

  1. Line 193. A better choice or word for "Besides" would be "Furthermore".

We took the advice and used “Furthermore” instead of “Besides”

  1. Lines 213-214. This is an important point that the subterranean environment would be suitable habitat for the fungus, and losses would go undetected

Thank you. We are uncertain whether we should elaborate on this further.

Reviewer 2 Report

The current manuscript is predicting the invasion risk of the fungal pathogen Batrachochytrium salamandrivorans of populations of threatened Mediterranean salamanders Lyciasalamandra spp. using a combination of field experimentation and controlled laboratory trials.

Noteworthy the studied salamanders are endangered, therefore the study by Li et al. has direct impact on species conservation measures. Still conservation biologists need first hand information on how to estimate fungal pathogen impacts on salamanders. To the best of my knowledge the work presents also first hand pioneering data on the thermal ecology of Lyciasalamandra (albeit given the species limited range and the regional character of the study).

The presentation of the MS has been carried out very well, the reader gets a good insight into the theme, the results are absolute convincing, the discussion is adequate and covers all the relevant aspects. The sample size is absolute adequate.

Generally I do not see any obstacles why the current topic is not interesting for the audience of JOURNAL OF FUNGI. The study by Li et al. certainly will attract readers from various research fields e.g., mycologists, ecologists, conservation biologists and of course herpetologists.

Minor comments

Material & methods
P3, l109ff please add information on the tank size in which the field/laboratory experiments were carried out

References
P8, l277 set species name into italics
P9, l326ff set genus and species name into italics
P9ff set all species names into italics!

Bests

Alex Kupfer

Author Response

  1. P3, l109ff please add information on the tank size in which the field/laboratory experiments were carried out.

We thank the reviewer for this suggestion. We added the container size to line 112.

  1. References
    P8, l277 set species name into italics
    P9, l326ff set genus and species name into italics
    P9ff set all species names into italics!

Thank you. All references have been checked and amended.